# Non-Association of Driver Alterations in PTEN with Differential Gene Expression and Gene Methylation in IDH1 Wildtype Glioblastomas

**DOI:** 10.3390/brainsci13020186

**Published:** 2023-01-23

**Authors:** Mrinmay Kumar Mallik, Kaushik Majumdar, Shiraz Mujtaba

**Affiliations:** 1Department of Laboratory Medicine, Mubarak Al Kabeer Hospital, P.O. Box 43787, Hawally 32052, Kuwait; 2Department of Pathology, Al Sabah Hospital, P.O. Box 4078, Shuwaikh Industrial 70051, Kuwait; 3Department of Biology, City University of New York, Brooklyn, NY 11225, USA

**Keywords:** PTEN, driver alterations, IDH1 wildtype glioblastoma, Cbioportal, transcriptome, methylome, cancer genomics, gene ontology, TCGA

## Abstract

During oncogenesis, alterations in driver genes called driver alterations (DAs) modulate the transcriptome, methylome and proteome through oncogenic signaling pathways. These modulatory effects of any DA may be analyzed by examining differentially expressed mRNAs (DEMs), differentially methylated genes (DMGs) and differentially expressed proteins (DEPs) between tumor samples with and without that DA. We aimed to analyze these modulations with 12 common driver genes in Isocitrate Dehydrogenase 1 wildtype glioblastomas (IDH1-W-GBs). Using Cbioportal, groups of tumor samples with and without DAs in these 12 genes were generated from the IDH1-W-GBs available from “The Cancer Genomics Atlas Firehose Legacy Study Group” (TCGA-FL-SG) on Glioblastomas (GBs). For all 12 genes, samples with and without DAs were compared for DEMs, DMGs and DEPs. We found that DAs in *PTEN* were unassociated with any DEM or DMG in contrast to DAs in all other drivers, which were associated with several DEMs and DMGs. This contrasting *PTEN*-related property of being unassociated with differential gene expression or methylation in IDH1-W-GBs was unaffected by concurrent DAs in other common drivers or by the types of DAs affecting *PTEN*. From the lists of DEMs and DMGs associated with some common drivers other than PTEN, enriched gene ontology terms and insights into the co-regulatory effects of these drivers on the transcriptome were obtained. The findings from this study can improve our understanding of the molecular mechanisms underlying gliomagenesis with potential therapeutic benefits.

## 1. Introduction

Accumulating alterations in cancer driver genes like mutations, copy number alterations (CNAs) and translocations play a central role during oncogenesis [1]. Mutations steering tumorigenesis are referred to as driver mutations [2]. The other mutations, which do not have any apparent effect on cancer progression, are called passenger mutations [2], although some studies suggest that passenger mutations may accelerate tumor progression [3]. The term driver alteration (DA) is defined as a change in a driver gene that has the potential to drive neoplastic processes [4,5]. A driver gene can undergo many types of DAs [6,7]. For example, in glioblastomas (GBs), *EGFR* is altered by different types of mutations or amplifications [6]. Although the type of DA may significantly affect a tumor’s phenotype [8,9,10,11], its effect is mediated by either the enhancement or attenuation of the function associated with the driver gene [12,13]. DAs dictate tumor phenotype by modulating signaling pathways [14,15] that lead to changes in the transcriptome [16,17], methylome [18,19] and proteome [20,21]. Alterations in mRNA expression, DNA methylation and protein expression are major representatives of such changes [16,17,18,19,20,21].

A tumor is susceptible to undergoing DAs in multiple genes [6,7]. These DAs influence their transcriptional, DNA methylation and proteomic profiles [14,22]. Many tumors such as breast cancers and GBs have been classified into clinically and therapeutically relevant subclasses based on their gene expression [23,24] and DNA methylation profiles [25,26]. These profiles represent the molecular characteristics of the predominant cell population [27,28]. However, due to tumor heterogeneity, neoplasms consist of several molecularly heterogeneous cell populations of variable sizes that may possess different types of DAs [27,28]. Determining differences in gene expression, gene methylation and protein expression between these populations can help with understanding the differences in their biological properties.

Establishing the contribution of a specific DA toward a tumor phenotype remains a challenge. By comparing groups of cases with and without DAs in a particular driver for differentially expressed mRNAs (DEM), differentially methylated genes (DMGs) and differentially expressed proteins (DEPs), one might obtain crucial insights into ways by which that driver affects the transcriptome, methylome and proteome in these tumors. Toward such goals, studies using DEMs [29,30,31,32], DMGs [32,33,34] and DEPs [35,36] have been published. To understand their biological significance, such lists of DEMs, DMGs and DEPs can be further analyzed to find enriched gene ontology (GO) terms under the categories, “molecular functions (MFs)” “biological processes (BPs)” and “cellular components (CCs)” [37]. Enriched “biological pathways (BPWs)” may also be analyzed for such lists [38]. The **D**atabase for **A**nnotation, **V**isualization and **I**ntegrated **D**iscovery (**DAVID**) provides a comprehensive set of functional annotation tools for investigators to understand the biological meaning of lists of genes [39]. The functional annotation tools in DAVID enable users to perform GO analyses for enriched GO terms and BPWs [39]. Apart from these qualitative analyses, a quantitative analysis may also be biologically and prognostically meaningful. For instance, when different grades of gliomas were compared with normal brain tissue for DEMs, the numbers of DEMs were highest in CNS grade 4 tumors, followed by grade 3 and grade 2 astrocytoma [40]. The lists of DEMs, DMGs and DEPs may be obtained from experimental data derived from wet laboratory experiments or publicly available databases [40]. The Cancer Genome Atlas (TCGA) program has characterized over 20,000 primary cancers in 33 cancer types, generating over 2.5 petabytes of publicly available genomic, epigenomic, transcriptomic and proteomic data [41,42]. The TCGA program has increased our understanding of the molecular constitution of cancers [1,42,43], led to advances in health science technologies [43] and changed the way cancer patients are treated in clinics [44]. However, the systematic analyses of such large datasets can be overwhelming for researchers without specialized bioinformatic skills.

CBioportal is a web platform (cbioportal.org) that allows the analyses of large-scale cancer genomics data collated from resources such as TCGA, The International Cancer Genome Consortium and public sequencing studies [45]. Researchers can select datasets related to a certain cancer of interest and perform virtual analyses for correlating clinical data, epidemiological characteristics, survival figures, histopathology images and molecular profiles. Cbioportal is equipped with several user-friendly tools. One of the tools allows users to create virtual groups of patients based on a specific characteristic or a combination of characteristics associated with a tumor type. For example, it is possible to create groups of patients with or without DAs in selected genes. For this, Cbioportal utilizes an OncoQuery language-powered search tool [45]. Several studies in the recent past have used the Cbioportal platform to understand the biology and behavior of specific types of cancers and molecules [46,47,48,49]. Cbioportal has been used to perform multiomics analyses of the effects of bone morphometric proteins in various cancers [46], to identify keys genes and pathways in endometrial cancers [47], to analyze the role of STAT3 in different cancers [48] and to analyze the role of SUMOylation-related molecules as prognostic markers in glioblastomas [49].

Despite major advances in the understanding of its biological properties and developments in therapeutic modalities, glioblastoma, the most common primary brain malignancy, has continued to be plagued by extremely poor survival [50,51]. GBs with mutations in the *IDH1* gene (IDH1-M-GB), presently designated as Astrocytoma, IDH mutant, CNS WHO Grade 4 [52], which account for approximately 5% of GBs [53,54], are molecularly, biologically and behaviorally distinct from those without those mutations. The latter are called IDH1 wildtype glioblastomas (IDH1-W- GB) [52]. GBs are also classified into distinct molecular classes that have prognostic and therapeutic connotations based on gene expression profiles [55,56]. Although these classes are created based on gene expression profiles, the prevalence of altered driver genes differs significantly between them [55,56].

This study uses the CBioportal platform to understand the impact of DAs (individually or as combinations) on the transcriptomes, methylome and proteomes of primary IDH1-W-GBs. Among the common DAs that drive gliomagenesis, those in *PTEN* have been extensively studied [57,58,59]. However, in this study, we show a previously unknown (to the best of our knowledge) property associated with *PTEN* in IDH1-W-GB. DAs in *PTEN* are not associated with any DEMs or DMGs. This fact became evident when groups of IDH1-W-GBs with and without DAs in *PTEN* were compared for DEMs and DMGs. *PTEN’s* lack of association with DEMs and DMGs contrasts with other common DAs because they are associated with several DEMs and DMGs. Additionally, through the detection of enriched MFs, BPs, CCs and BPWs from lists of DEMs and DMGs associated with some other common DAs in IDH1-W-GBs, such as *CDKN2A*, *EGFR* and *TP53*, we highlight the biological relevance of these drivers in contributing toward the tumor phenotype. Furthermore, by determining the concurrently upregulated and/or downregulated genes associated with these drivers, we uncover some possible co-regulatory effects of these drivers on the transcriptome.

Exploring the molecular mechanisms underlying these findings can broaden our understanding of the influence of DAs on the multiomics landscape in these tumors and can consequently be translated into developing better treatment strategies.

## 2. Materials and Methods

### 2.1. Access to Data on Cbioportal

On Cbioportal [45], the data were extracted from The Cancer Genome Atlas Firehose Legacy Study Group (TCGA-FL-SG) on glioblastomas (GBs) and the Brain Tumor PDX (Mayo clinic 2019) on glioblastomas, which is a brain tumor patient-derived xenograft resource. The primary analysis was performed on the TCGA-FL-SG on GBs. Furthermore, the Brain Tumor PDX (Mayo Clinic 2019) on GBs was used for secondary analyses to evaluate the association of driver alterations in *PTEN* because of the findings from the primary analyses (Appendix A: Appendix A). Three additional analyses were performed. The first of these looked for correlations among gene expression, gene methylation and protein expression. The second one was performed to determine enriched GO terms and BPWs amongst the list of DEMs and DMGs associated with DAs in some common driver genes. The third looked for DEMs and DMGs that were common between the lists of DEMs and DMGs found in association with DAs in some of the common drivers.

### 2.2. Primary Analyses of TCGA-FL-SG on GBs

Figure 1 illustrates the pipeline for the primary analyses. Section 2.2.1, Section 2.2.2, Section 2.2.3, Section 2.2.4, Section 2.2.5, Section 2.2.6, Section 2.2.7 and Section 2.2.8 listed below describe the steps involved in the analysis of the TCGA-FL-SG on GB.

#### 2.2.1. Creation of Three Subsets of Cases Called Core Groups for Three Different Types of Analyses

From the TCGA-FL-SG on GBs, the recurrent samples were excluded to focus the study on primary GBs only (Appendix A: from v to vii). Subsequently, three initial groups of cases were generated from the primary GBs. Then, three core groups of cases were derived from the above three initial groups to study the associations of DAs with the transcriptome, methylome and proteome, respectively. Every case in the TCGA-FL-SG on GBs does not exhibit a complete profile of all tests to allow a concurrent evaluation of the transcriptome, methylome and proteome for the entire study group. Therefore, it was necessary to create three separate groups of cases based on the availability of the required individual test profiles.

The first initial group, created for the analyses of the transcriptome, comprised cases with concurrently available profiles of CNAs, mutations and mRNA expression, which were performed on RNA sequencing platform V2 and analyzed with the RNA sequencing by expectation maximization (RSEM) software package (Appendix A of Appendix A). The second initial group, for the analyses of the methylome, comprised cases with concurrently available test profiles of CNAs, mutations and DNA methylation performed on the Illumina HM450 platform (Appendix A of Appendix A). The third initial group for analyses of the proteome comprised cases with concurrently available test profiles of CNAs, mutations and protein expression on a Reverse Phase Protein Array (RPPA) (Appendix A of Appendix A).

Next, from each of these three initial groups, IDH1-M-GBs were excluded (Appendix A of Appendix A). The remaining IDH1-W-GBs from each constituted three new groups of cases, which were named core group 1 (C1), core group 2 (C2) and core group 3 (C3) for the analyses of the impact of DAs on the transcriptome, methylome and proteome, respectively.

#### 2.2.2. Selection of Driver Genes for Analyses

Cbioportal uses the MutSig algorithm [60] to identify genes that are mutated more often than expected by chance based upon the assessment of background mutation processes and hence have the potential to be driver genes. Similarly, the GISTIC algorithm [61] is used to identify genes with likely driver CNAs. Cbioportal also shows the prevalence of altered driver genes within a group of cases [45]. The prevalence of altered driver genes in the three core groups was recorded. Previously well-studied glioblastoma-related driver genes with a prevalence of DAs of 5% or more in any of the above-mentioned core groups were selected for further analyses. Cbioportal confirmed the identities of these genes as potential drivers in IDH1-W-GBs in agreement with the above-mentioned MutSig and GISTIC algorithms.

A 5% prevalence cut-off was selected because the prevalence of IDH1 mutations in the entire study group was 5%, and IDH1 mutant cases were excluded from the analyses, as previously mentioned.

The above process led to the selection of 12 driver genes, i.e., *EGFR, CDKN2A, PTEN, TP53, CDK4, MDM2, MDM4, NF1, PDGFRA, PIK3CA, PIK3R1* and *RB1*. Notably, because all cases with DAs in *CDKN2B* exhibited DAs in *CDKN2A*, the DAs in *CDKN2B* were not separately analyzed.

#### 2.2.3. Creating Groups with and without DAs in a Specific Gene

For this study, the phrase “driver alteration” merges driver mutations and CNAs. This approach is a convenient one to group cases with or without DAs in any driver gene on Cbioportal. With its user interface, one can select groups of cases with DAs in any driver gene using appropriate queries. Cases that constitute such a group comprise all cases with any DA in that driver gene, irrespective of the type of DA. Similarly, it is possible to create a group without any DA in that driver gene using a slight modification. Using this tool, two groups of cases were created from each of the 12 driver genes. One group comprised cases with DAs, whereas the other comprised cases without DAs in that particular driver gene. Thus, 24 groups of cases (12 pairs each for each gene) were created (Appendix A of Appendix A).

#### 2.2.4. Associations between DAs in Individual Genes with DEMs, DMGs and DEPs

The prevalence of cases with DAs in each of the 12 driver genes in groups C1, C2 and C3 was recorded. DEMs, DMGs and DEPs between groups of cases with or without DAs in individual drivers represent changes in the transcriptome, methylome and proteome associated with such drivers [31,32,33,34,35,36,37,38]. To study these changes, the following methods were replicated in groups C1, C2 and C3.

The C1 group was analyzed to study the impact of DAs on the transcriptome. Toward this goal, groups of cases with and without DAs in the 12 genes were compared for DEMs. The counts of mRNAs that are upregulated and downregulated to a statistically significant extent (as estimated by Cbioportal using a q value cut-off of 0.05) and the sum of these DEMs were recorded.

The C2 group was analyzed to study the impact of DAs on the methylome. Toward this goal, groups of cases with and without DAs in the 12 genes were compared for DMGs. The counts of genes that are hypermethylated and hypermethylated to a statistically significant extent (as estimated by Cbioportal using a q value cut-off of 0.05) and the sum of these DMGs were recorded.

The C3 group was analyzed to study the impact of DAs on the proteome. Toward this goal, groups of cases with and without DAs in the 12 genes were compared for DEPs. The counts of proteins that are upregulated and downregulated to a statistically significant extent (as estimated by Cbioportal using a q value cut-off of 0.05) and the sum of these DEPs were recorded.

Appendix A of Appendix A summarizes the above protocols.

To determine if any significant correlation exists between the prevalence of genes and DAs in C1, the counts of upregulated mRNAs, the counts of downregulated mRNAs or the counts of total differentially expressed mRNAs associated with them, Spearman’s Rank Correlation Coefficient was used with two non-parametric data sets at a time. Similar analyses were performed for any correlation between the prevalence of genes and DAs in C2, the counts of hypermethylated genes, the counts of hypomethylated genes or the total differentially methylated genes associated with them. However, these analyses were not performed on C3 because there were low counts for DEPs (shown in the Section 3).

#### 2.2.5. Associations between DAs in Individual Genes with Clinical Features and Survival

The associations between DAs in individual driver genes with clinical features and survival figures were analyzed using the inbuilt tools in Cbioportal (Appendix A of Appendix A).

The findings from the above analyses (Section 2.2.1, Section 2.2.2, Section 2.2.3, Section 2.2.4 and Section 2.2.5) revealed that DAs in *PTEN* are unassociated with DEMs and DMGs in IDH1-W-GBs, unlike DAs in other genes (shown in the Section 3). Therefore, further analyses were focused on studying how other variables and conditions might affect this non-association of DAs in PTEN with DEMs and DMGs.

#### 2.2.6. Association between PTEN and Other Analyzed Genes

Noting that the impacts of *PTEN* on the transcriptome and methylome were different from those of other drivers, it was necessary to determine any significant association between DAs in *PTEN* and the other 11 analyzed genes. Chi-squared tests (or Fisher’s exact tests, as the situation required) for a 2 × 2 contingency table were carried out, with the calculations of odds ratios (ORs) and 95% confidence intervals (95% CIs) using data sets of 2 gene pairs at a time. These analyses were separately carried out for groups C1, C2, and C3.

#### 2.2.7. Influence of Concurrent DAs in Other Drivers on the Non-Association of DAs in PTEN with DEMs and DMGs

Groups of cases with combinations of DAs can be generated using tools in Cbioportal (Appendix A of Appendix A). Using these tools, appropriate groups with combinations of DAs in different drivers were created to determine if the presence or absence of concurrent DAs in other drivers influence the *PTEN*-related property of non-association with any DEM or DMG. A total of 4 of the 11 other driver genes were selected for this purpose, which were *EGFR*, *CDKN2A*, *TP53* and *RB1*. The reasons for selecting these four drivers were as follows. First, DAs in *CDKN2A*, *EGFR*, *PTEN* and *TP53* were the most prevalent DAs in C1 and C2, with 93.13% and 95.34% of the cases in C1 and C2 showing DAs in at least one of these four genes. These latter findings are described in the Section 3. Second, *RB1* was selected because DAs in *RB1* were the only DAs that showed a statistically significant positive association with DAs in *PTEN*. These findings are described in the Section 3. The groups that were created with appropriate combinations of DAs to evaluate the influence of the presence or absence of concurrent DAs in these four genes on *PTEN* are illustrated in Table 1.

#### 2.2.8. Associations between Types of DA in PTEN with DEMs and DMGs

From cases exhibiting DAs in *PTEN*, separate groups of cases with different types of DAs in the *PTEN* gene were created. These groups were compared for DEMs and DMGs in C1 and C2 (Appendix A of Appendix A).

### 2.3. Secondary Analyses of Other Datasets Pertaining to DAs in PTEN

Secondary analyses were performed on another analysis platform and another dataset of IDH1-W-GBs to determine if the *PTEN* property of non-association with DEM or DMG is replicated.

(i).From the TCGA-FL-SG on GBs, using selection criteria such as those used to create the C1 group, a separate dataset was created using cases profiled on the U133 microarray platform (instead of the RNA sequencing v2 platform with RSEM, on which the primary analysis was performed). Groups of cases with and without DAs in *PTEN* were compared for DEMs.(ii).The IDH1-W-GBs in the Brain Tumor PDX (Mayo Clinic 2019) dataset were analyzed to determine the association of the DAs in *PTEN* with DEMs (in this dataset, the gene expression profile was available on the RNA sequencing v2 platform) and DMGs (in this dataset, the methylation test profile was available on the Illumina EPIC-850k platform) [45].

### 2.4. Additional Analyses

#### 2.4.1. Analyses for Correlation among DEMs, DMGs and DEPs

A separate core group named C4 was generated from the TCGA-FL-SG on GBs. It comprised cases in which test profiles for CNAs, mutations, mRNA expressions, methylations and protein expressions (RPPA) were concurrently available. Following the methodology used in the primary analyses, lists of DEMs, DMGs and DEPs were created for DAs in *EGFR*, *CDKN2A* and *TP53*. These lists were compared to look for any correlations among DEMs, DMGs and DEPs. Figure 2 shows the workflow of the above analyses in the C4 group.

#### 2.4.2. Analyses for Enriched GO Terms and Pathways among DEMs and DMGs Associated with DAs in CDKN2A, EGFR and TP53

The lists of DEMs and DMGs found in association with DAs in *CDKN2A*, *EGFR* and *PTEN* are shown in Appendix A, respectively. These DEMs and DMGs were analyzed for enriched MFs, BPs, CCs and BPWs. The functional annotation tools incorporated in DAVID were used for these enrichment analyses, following the recommended steps [39]. These are detailed in Appendix A in Appendix A. A maximum of ten of the most significantly enriched GO terms and pathways were tabulated.

#### 2.4.3. Analyses of DEMs and DMGs That Are Common between the DEMs and DMGs Associated with DAs in CDKN2A, EGFR and TP53

All the lists of upregulated and downregulated DEMs (shown in Appendix A) in association with DAs in *CDKN2A*, *EGFR* and *TP53* were compared in pairs to determine the common DEMs. Thus, six lists (found in association with upregulated and downregulated DEMs for the above three drivers) were compared. From some of these comparisons, we found common DEMs. In these situations, we created separate lists comprising these common DEMs. The latter lists were analyzed using DAVID to find enriched GO terms and biological pathways.

Similarly, the lists of DMGs (shown in Appendix A) for the same drivers were compared to determine the common DMGs. However, only one common DMG was found (shown in the Section 3), so no list was generated.

## 3. Results

This study aimed to determine how DAs might drive gliomagenesis by examining the transcriptome, methylome and proteome of IDH1-W-GBs using Cbioportal. The hypothesis was based on the notion that each DA uniquely impacts the transcriptome, methylome and proteome, which ultimately drives tumorigenesis and determines the phenotype. Toward this goal, patient samples were grouped according to the presence or absence of DAs in specific genes. The primary analyses were performed on the data extracted from the TCGA-FL-SG on GBs.

Secondary analyses were performed to confirm the findings from the primary analyses. These secondary analyses were performed on data extracted from an analysis platform and GB dataset that were different from those used in the primary analyses.

Additional analyses were performed to find correlations between gene expression and gene methylation, to find enriched GO terms and to find common DEMs and DMGs associated with some common drivers.

The results are described below.

### 3.1. Primary Analyses on the TCGA-FL-SG on GBs

#### 3.1.1. Data Resources and Data Preparation

The TCGA-FL-SG dataset on GBs comprises 619 samples from 606 patients. Of these 619 samples, 13 are recurrences, and 606 are from primary samples. From these primary GB samples, three core groups of IDH1-W-GB cases, namely C1, C2, and C3, were generated for analyzing the associations between DAs in 12 driver genes with DEMs, DMGs and DEPs. The C1, C2, and C3 groups comprised 129, 102 and 132 cases, respectively.

#### 3.1.2. Associations between Individual DAs and DEMs

Table 2 shows the prevalence of 12 driver genes with DAs amongst the 129 cases in the C1 group, along with the counts of significantly up and downregulated mRNAs and their sum (i.e., total DEMs) associated with these DAs. *CDKN2A, EGFR*, *PTEN* and *TP53* were the four most prevalent driver genes in the C1 group. Furthermore, it was noted that DAs in at least one of these four drivers were present in 93.13% of 129 cases in the C1 group. The DAs in all driver genes were associated with several DEMs except those involving *PTEN*. It was noted that not a single mRNA was either up or downregulated in association with the DAs in *PTEN*. The list of the DEMs is provided in Appendix A.

The Bivariate analyses using Spearman’s rank correlation showed no significant correlation between the prevalence of DAs in the analyzed genes and their corresponding differentially expressed mRNA counts, including upregulated, downregulated and total DEMs, as shown in Appendix A.

#### 3.1.3. Associations between Individual DAs and DMGs

Table 3 shows the prevalence of 12 driver genes with DAs amongst the 102 cases in C2 group, along with the counts of significantly hyper and hypomethylated genes and their sum (i.e., total DMGs) associated with these DAs. *CDKN2A, EGFR*, *PTEN* and *TP53* were the most prevalent driver genes in the C2 group. Of cases in the C2 group, 95.34% showed DAs in at least one of these four drivers. Genes *NF1, RB1* and *MDM4* were the ones that were found to be associated with the three highest total numbers of DMGs. All driver genes were associated with DMGs except *PTEN* and *PIK3R1*. DAs in neither *PTEN* nor *PIK3R1* led to any DMG. The lists of the DMGs are provided in Appendix A.

The Bivariate analyses using Spearman’s rank correlation showed no significant correlation between the prevalence of DAs in the 12 analyzed genes and their corresponding differentially methylated gene counts, including hypermethylated, hypomethylated and the total, as shown in Appendix A.

#### 3.1.4. Associations between Individual DAs and DEPs

Table 4 shows the prevalence of 12 driver genes with DAs amongst the 132 cases in the C3 group, along with the counts of significantly overexpressed and underexpressed proteins and their sum (i.e., total DEPs) associated with these DAs. The four most prevalent genes with DAs in the C3 group were *CDKN2A, EGFR, PTEN* and *TP53*. The DAs associated with *EGFR* had 14 proteins that were differentially expressed, which was the most among the 12 driver genes that were analyzed. DAs associated with several of these genes showed no associated DEP. It was noted that DAs in *PTEN* were associated with the differential expression of multiple AKT isoforms. Appendix A shows the list of DEPs.

#### 3.1.5. Statistical Analyses for Association between PTEN and other Genes

In the C1 group, the only driver gene with which *PTEN* was positively associated was *RB1*. In this group (*n* = 129), the odds ratio was 4.34 (Fisher’s exact test, 95% confidence interval: 1.28–14.71) for *PTEN* and *RB1*. No other significant association could be demonstrated between *PTEN* and any of the other 11 genes analyzed in the C1, C2 or C3 groups, as shown in Appendix A.

#### 3.1.6. Associations between Individual DAs with Clinical Features and Survival

Groups C1, C2 and C3 were analyzed for their associations between individual DAs with clinical features and survival. DAs in none of the driver genes were associated with any clinical attributes, including race, age at diagnosis and performance status. Moreover, none of them individually had any association with overall survival or progression-free survival.

#### 3.1.7. Influence of Concurrent DAs on the Non-Association of DAs in PTEN with DEMs and DMGs

Table 5 shows the influence of the presence or absence of concurrent DAs in *EGFR, CDKN2A, TP53* and *RB1* with DAs in *PTEN* on the *PTEN* property of non-association with differential gene expression and gene methylation. It is evident that, when cases with and without DAs in *PTEN* are compared, in the presence or absence of DAs in *EGFR*, *CDKN2A* and *TP53*, no DEMs or DMGs were noted. Only CPS4L mRNA was downregulated in the presence of concurrent DAs in *RB1*.

#### 3.1.8. Associations between Types of DAs in PTEN with DEMs and DMGs

Out of the 52 cases with DAs in *PTEN* in the C1 group, 19 showed only deep deletions, one showed both deep deletions and truncating mutations, 18 showed only truncating mutations, 3 showed only splice mutations and 16 showed only missense mutations. No DEM was noted between any of the groups when they were compared with each other.

Out of the 32 cases with DAs in *PTEN* in the C2 group, 7 showed deep deletions, 13 showed truncating mutations, 3 showed splice mutations and 9 showed missense mutations. No DMG was noted between any of the groups when they were compared with each other.

### 3.2. Secondary Analyses on the Association between DAs in PTEN on Another Analysis Platform and Another Data Resource on GBs

The association of DAs in *PTEN* with the differential expression of mRNA using the mRNA expression profile on the U133 platform for IDH1-W-GBs in the TCGA-FL-SG showed no DEMs.

A total of 57 cases of IDH1-W-GBs from Brain Tumor PDX (Mayo Clinic 2019) were analyzed for the association of DAs in *PTEN* with DEMs and DMGs. These comprised 20 samples with DAs and 37 without DAs in *PTEN*. Not a single DEM was associated with DAs in *PTEN* in these cases. A total of 65 samples were analyzed for the association of DAs in *PTEN* with DMGs. These comprised 22 samples with DAs and 43 without DAs in *PTEN.* No DMG was detected.

### 3.3. Additional Analyses

#### 3.3.1. Analyses for Correlation between DEMs, DMGs and DEPs

There were 45 cases in the TCGA-FL-SG on GBs where the results of CNAs, mutations, mRNA expressions, gene methylations and protein expressions were concurrently available to attempt correlation analyses between the associated DEMs, DMGs and DEPs. This comprised the C4 group. Out of these 45 cases, 12 (26.67%) had DAs in *PTEN,* 28 (62.22%) had DAs in *CDKN2A*, 20 (44.44%) had DAs in *EGFR* and 8 (17.78%) had DAs in *TP53*. There were no DEMs or DMGs associated with DAs in *PTEN*. There were three DEPs (all were underexpressed AKT isoforms) associated with DAs in *PTEN*. There were no DEMs associated with DAs in *CDKN2A*. However, there were five hypomethylated genes associated with DAs in *CDKN2A*. On the other hand, no DEP was associated with DAs in *CDKN2A*. There were no DEMs or DMGs associated with DAs in *EGFR*. There were four DEPs (all overexpressed EGFR isoforms) associated with DAs in *EGFR*. There were 518 DEMs associated with DAs in *TP53,* which comprised 291 upregulated and 227 downregulated genes. There were 23 DMGs associated with DAs in *TP53,* which comprised 5 hypermethylated and 18 hypomethylated genes. The only significant correlation that could be detected was in association with DAs in *TP53*. This was the *VWC* gene, which was found to be hypomethylated in association with DAs in *TP53*, and its expression was upregulated in association with DAs in *TP53*. The lists of DEMs, DMGs and DEPs are shown in Appendix A.

#### 3.3.2. Analyses for Enriched GO Terms and Pathways among DEMs and DMGs Associated with DAs in CDKN2A, EGFR and TP53

Appendix A shows the various enriched GO terms and pathways for DEMs in association with DAs in *CDKN2A*, *EGFR* and *TP53*. Some of the contextually relevant findings are as follows.

Among the mRNAs upregulated in association with DAs in *CDKN2A*, ‘inactivation of the MAPK pathway’ is an enriched BP. Endoplasmic reticulum and related structures are prominent amongst enriched CCs, and ‘aberrant regulation of the G1/S phase in cancer due to RB1 defects’ stands out as an enriched BPW. For mRNAs downregulated in association with DAs in *CDKN2A,* those of note were ‘glutamate-receptor-related functions and processes’ and ‘ion-channel-related functions and processes’ amongst enriched MFs and BPs. In addition, ‘regulation of the G-protein-coupled receptor protein signaling pathway’ was among the MFs. The BPWs of note include those related to ‘G-protein-coupled receptor signaling’, ‘Cyclin-D-associated events in G1′ and ‘oncogene-induced senescence’.

Among the mRNAs upregulated in association with DAs in *EGFR*, ‘RNA polymerase II’-related MFs are frequently enriched, and this is also noted in the enriched BPs. Enriched CCs in this context are distributed widely within different cell compartments. Among the enriched BPWs, there were ‘interferon alpha/beta signaling’, ‘interferon gamma signaling’ and ‘RNA polymerase II transcription’. Among the mRNAs downregulated in association with DAs in *EGFR*, the noteworthy enriched MFs were ‘growth factor binding’ and ‘BMP receptor activity’, and ‘regulation of Wnt signaling’ and ‘BMP signaling’ were among those in the enriched BPs. Among the enriched BPWs, ‘membrane trafficking’ and ‘negative regulation of TCF-dependent signaling by WNT ligand antagonists’ were noteworthy.

Among the mRNAs upregulated in association with DAs in *TP53*, some of the enriched MFs were related to the binding of nucleic acid, proteins, chromatin and ATPase, whereas ‘cell division’, ‘cell cycle’, ‘chromatin response to DNA damage’ and ‘DNA helicase activity’ were some of the enriched BPs. Most of the enriched CC terms were related to the nucleus, whereas the enriched BPWs included ‘cell cycle, ‘gene expression’, ‘metabolism of RNA’ and ‘transcriptional regulation of TP53′. Among the mRNAs downregulated in association with DAs in *TP53,* the enriched MFs of note included protein, integrin and ion channel binding and ‘oxidoreductase activity’. ‘Intrinsic apoptotic signaling pathway in response to DNA damage by p53 class mediator’, ‘cell cycle’ and ‘positive regulation of endothelial cell migration’ were some of the noteworthy BPs. The enriched CCs were mostly related to the mitochondria, Golgi apparatus and endoplasmic reticulum. The enriched BPWs included those related to ‘respiratory electron transport chain’, ‘Citric acid cycle’ and ‘TNFR2 non-canonical NFKB pathway’.

Appendix A shows the various enriched GO terms and pathways for the DMGs associated with DAs in *CDKN2A*, *EGFR* and *TP53*. There was no enriched term in relation to hyper or hypomethylated genes associated with DAs in *CDKN2A*, in relation to hypomethylated genes associated with DAs in *EGFR* and in relation to hypermethylated genes associated with DAs in *TP53*, because few DMGs were associated with these DAs. Among the hypermethylated genes associated with DAs in EGFR, one noteworthy MF that was seen to be enriched was ‘receptor tyrosine kinase activity’, and ‘positive regulation of cell division’, ‘PI3 Kinase signaling’ and ‘neural crest migration’ were some of the noteworthy BPs. The enriched CCs were mostly related to the plasma membrane. The only enriched BPW was ‘signal transduction’. Among the hypomethylated genes associated with DAs in *TP53*, the only enriched MF was ‘PDZ domain binding’. The enriched BPs included ‘regulation of synaptic vesicle exocytosis’, ‘neurotransmitter receptor localization of postsynaptic specialization membrane’ and ‘central nervous system development’.

#### 3.3.3. Analyses of the DEMs and DMGs That Are Common between the DEMs and DMGs Associated with DAs in CDKN2A, EGFR and TP53

We found 12 DEMs that were upregulated, in association with DAs in both *CDKN2A* and *EGFR*, and 25 mRNAs that were downregulated in association with DAs in both *CDKN2A* and *EGFR.* None of the DEMs upregulated in association with DAs in *CDKN2A* were downregulated in association with DAs in *EGFR* or vice versa.

No DEM upregulated in association with DAs in *CDKN2A* or *EGFR* was found to be upregulated in association with DAs in *TP53*. Similarly, no DEM downregulated in association with DAs in *CDKN2A* or *TP53* was found to be downregulated in association with DAs in *TP53*.

However, there were 14 DEMs that were upregulated in association with DAs in *CDKN2A* but that were downregulated in association with DAs in *TP53*, and there were 55 DEMs that were upregulated in association with DAs in *CDKN2A* but that were downregulated in association with DAs in *TP53*. Furthermore, there were 39 DEMs that were upregulated in association with DAs in *EGFR* but that were downregulated in association with DAs in *TP53*, and there were 84 DEMs that were upregulated in association with DAs in *CDKN2A* but that were downregulated in association with DAs in *TP53*.

To summarize, DAs in CDKN2A and EGFR were associated with some commonly upregulated and downregulated DEMs. However, DAs in *CDKN2A* (and in *EGFR*) were associated with opposing sets of upregulated and downregulated DEMs compared to DAs in TP53.

The lists of common DEMs are shown in Appendix A. The enriched GO terms and molecular pathways in association with them are shown in Appendix A.

Among the lists of DMGs associated with DAs in *CDKN2A*, *EGFR* and *TP53*, no common DMGs were detected, except *PRDM15*. This gene was hypermethylated in association with DAs in *EGFR* and was hypomethylated with DAs in *TP53*.

## 4. Discussion

### 4.1. The Fundamental Concepts That Guided the Basic Study Design

Using Cbioportal, we analyzed the IDH1-W-GBs in the TCGA-FL-SG on GBs to determine how DAs might influence the transcriptome, methylome and proteome. Toward this end, we generated lists of DEMs, DMGs and DEPs associated with these DAs for further analyses. Different types of DAs in a driver gene might differently influence gene expression [17] and methylation [63]. Therefore, one might argue that analyzing them individually instead of merging them as DAs would have been ideal. However, this approach would have led to a couple of strategic problems. First, the number of cases in many groups with specific types of DAs would have been very small. For instance, out of the 75 cases in this study that featured DAs in *CDKN2A* in the C1 group, 73 were CNAs in the form of deep deletions, and only 2 were truncating mutations. Moreover, of the 38 cases with DAs in *TP53*, 2 were deep deletions, and the remaining were due to different types of mutations (these data are not separately presented in the Section 3). Second, analyzing cases with and without a specific type of DA in a particular gene would lead to the inclusion of cases with a different type of DA in that gene among the cases without the specific type of DA being analyzed. For example, if we created a group with amplifications in *EGFR*, the group without this type of DA would contain cases with mutations in *EGFR*. Therefore, the different types of DAs were merged into a single phrase: ‘DA’.

### 4.2. Selection of Datasets and Creation of User-Defined Groups

The TCGA-FL-SG on GBs, one of the largest publicly available datasets on GBs, was ideal for our study. However, to ensure accurate analyses, it was necessary to divide the cases into three groups for reasons that are elaborated in Section 2.2.1 in the Section 2. IDH1-M-GBs represent a much smaller and biologically distinct subset of GBs as compared to IDH1-W-GBs. They lack DAs in *EGFR* and *PTEN* and show a far lower prevalence of DAs in *CDKN2A* [53,54]. Thus, they were excluded. For primary analyses, the RNA sequencing platform was preferred over the microarray platform because RNA sequencing detects a significantly higher number of DEMs than that of microarrays [64]. Similarly, the Illumina HM450 platform covering 450,000 DNA methylation sites was chosen instead of the older Illumina HM27 platform, which covers 27,000 sites [65]. A cut-off prevalence threshold of 5% was chosen because IDH1-M-GBs with an approximate prevalence of 5% were excluded. Consequently, notable GB-related driver genes such as *CDK6, TERT* and *ATRX,* with a prevalence of 2.7%, 2.28% and 1.35% in our cases, were excluded from analyses. Incidentally, *PIK3R1* fell marginally short (4.65%) of the cut-off prevalence threshold (5%) in C2 but was nevertheless included because, in C1 and C3, its prevalence was above the threshold.

### 4.3. Associations of DAs in PTEN with Differential mRNA Expressions and Gene Methylations

*PTEN* is one of the most frequently affected and studied tumor suppressor genes, notably associated with malignancies such as GBs, melanomas and cancers of the breast, prostate and endometrium [62,66]. The PTEN protein negatively regulates the intracellular levels of phosphatidylinositol-3,4,5-trisphosphate (PI3K) to inhibit the AKT/PKB signaling pathway [67].

In a previous study on prostatic adenocarcinoma in a TCGA dataset, Sun et al. [68] reported 1736 differentially expressed genes associated with DAs in *PTEN*. However, in this study, *PTEN* stood out as the only one among the 12 analyzed driver genes whose DAs were unassociated with any DEM or DMG in IDH1-W-GBs, both in the TCGA-FL-SG on GBS and in the Brain Tumor PDX (Mayo clinic 2019) study group cases. This property remained unchanged between platforms used to study gene expression and gene methylation.

Driver genes often act epistatically [69], enhancing and attenuating each other’s effects. Therefore, we explored the associations between DAs in PTEN with those in other genes. Only the DAs in *RB1* were positively associated with DAs in *PTEN* in the C1 group. Because, on their own, DAs in RB1 were associated with several DEMs and DMGs, it is unlikely that RB1 contributed to the non-association of *PTEN* with DEMs and DMGs. At this point, it is also important to note that, in any of the groups, there was no statistically significant correlation between the prevalence of DAs and the counts of DEMs and DMGs associated with them.

Because tumor evolution occurs through the accumulation of driver alterations [70,71], we studied the influence of concurrent driver alterations on the PTEN-related property. For this, we selected four driver genes: *EGFR*, *CDKN2A*, *TP53* and *RB1.* In the C1 and in C2 groups, 93.13% and 95.34% of cases had DAs in at least *EGFR*, *CDKN2A*, *PTEN* or *TP53,* ensuring wide coverage. *RB1* was selected because, in C1, it was the only driver positively associated with *PTEN.* None of these concurrent DAs had any effect on the property of *PTEN* to be unassociated with DEM or DMG, with the sole exception of a downregulated gene in the presence of concurrent DAs in RB1. These analyses were performed on various combinations of two driver genes. At this point, it cannot be said if performing more extensive analyses with combinations of three or more drivers might reveal situations in which DAs in *PTEN* behave differently. Future studies might be able to address such queries.

Different types of DAs in the same gene may exhibit variable associations with gene expression and methylation [17,63]. However, our analyses did not show any association between types of DAs in *PTEN* with differential gene expression and genes. This implies that the property of interest associated with DAs in *PTEN* is independent of the type of DA in *PTEN*.

The role of the PI3K pathway in influencing the above phenomenon might be speculated upon. Three other drivers in the PI3K pathway, i.e., *PDGFRA, PI3KA* and *PIK3R1*, were associated with several DEMs or DMGs except for *PIK3R1*, which showed no DMG. However, from Table 2 and Table 3, it appears that the numbers of DEMs and DMGs associated with these three drivers are lower than those associated with other drivers.

Because of the above lack of association with DEMs and DMGs, it is not possible to garner any biological meaningfulness through the detection of enriched GO terms amongst these DEMs and DMGs, as any system biology study [72] would suggest. Nevertheless, one might be able to speculate upon some biologically relevant implications of this finding. For instance, mechanisms that allow DAs in *PTEN* to get established within the evolutionary pattern of the accumulation of mutations [70,71], without significantly altering their transcriptome and methylome, can be explored. These investigations might look into molecular readjustments involving other players in tumor biology, e.g., non-coding RNAs such as microRNAs [73,74]. Longitudinal and single-cell transcriptomic analyses [75] integrated with multiomic approaches [72,76] may also be used. Novel tools such as the centrality analysis of molecular networks [77] and machine learning [76] might also provide crucial insights. The implications of *PTEN*’s lack of association with DEMs and DMGs for tumor heterogeneity [28] is another perspective of interest. For example, two populations of cells in a IDH1-W-GB, one with and one without DAs in *PTEN,* would be similar in their transcriptome and methylome profiles. In situations where one of these populations is more sensitive to a particular therapeutic maneuver, the less sensitive population might get established as the dominant population after therapy without altering its transcriptome or methylome after the abrogation of the more therapeutically sensitive group [78,79]. It is also imperative to note that our findings reflect the associations of DAs in *PTEN* with DAs in other genes during the evolution of IDH1-W-GBs. We have not looked into these associations of DAs by comparing tumor and normal brain tissue for DEMs, DMGs or DEPs. These issues might be addressed through future studies by comparing datasets from tumor samples and normal brain tissue samples using an approach similar to that used by Lucifero et al. [80].

### 4.4. Associations between DAs and Differential Protein Expression

The correlations between mRNA and protein expressions are generally poor [81]. However, Koussounadis et al. [82] showed a significant correlation between DEMs and DEPs in an ovarian carcinoma xenograft model. In our study, the numbers of DEMs and DMGs were considerably higher than those of DEPs. Thus, the proteomic profiles of cases differing in DAs in individual genes are more similar to each other than their transcriptomic and gene methylation profiles are.

### 4.5. Associations between DAs with Clinical Features and Survival Figures

For expected lines, none of the 12 driver genes showed any association with any clinical feature or survival figures. Gene-expression-based molecular classification of GBs has clinical, prognostic and therapeutic connotations [24]. However, none have been shown to have any associations with these parameters individually in IDH1-W- GBs. Meta-analyses of studies on gliomas (including all grades) have proposed that the expression of the EGFR protein [10] and mutations in *PTEN* [83] may have negative effects on overall survival, although their effects have not been evaluated individually in IDH1-W-GBs.

### 4.6. Associations among DEMs, DMGs and DEPs in a Single Group of Cases

Correlations among DEMs, DMGs and DEPs may yield interesting insights into coregulatory mechanisms within the multiomics landscape of tumors [32,75]. There were only 45 cases where all necessary test profiles were available to simultaneously detect DEMs, DMGs and DEPs. Although the number of cases was considerably lower in this group as compared to the three groups for which the primary analyses were performed, we decided to analyze this group to look for any correlation among gene expression, gene methylation and protein expression. However, this effort did not contribute significantly toward any noteworthy insights except for the *VWC2* gene, which was upregulated and hypomethylated in association with DAs in *TP53*. VWC2 (Von Willebrand factor C Domain containing 2), also called Brorin, is a bone morphogenetic protein antagonist which is possibly involved in neural function and development and may have a role in cell adhesion [84]. Interestingly, *EGFR2:VWC2* has been suggested to be a therapeutically targetable fusion in IDH1-W-GBs [85].

### 4.7. Determining the Biological Relevance of DEMs and DMGs Associated with DAs in CDKN2A, EGFR and TP53

Some studies on gliomas have used GO and pathway analyses as bioinformatic tools to determine the biological significance of lists of genes from enriched GO terms and pathways [86,87,88]. These lists comprise genes that are differentially expressed between normal brain samples and GBs [86], genes that are differentially expressed between different grades of gliomas [87] and genes associated with a hereditary predisposition to GBs [88]. However, to the best of our knowledge, there has not been any publication yet that has used GO or pathway enrichment analyses associated with various DAs in GBs. The lack of DEMs and DMGs precludes the possibility of subjecting DAs in PTEN to GO analyses. However, the list of DEMs and DMGs associated with some of the other DAs presented us with a unique opportunity to subject them to GO and pathway enrichment analyses using DAVID [39]. These analyses provided us with a fair number of interesting insights. The GO terms and pathways are shown in Appendix A for gene-expression-related and gene-methylation-related data. Some of the noteworthy terms we considered to be contextually relevant are separately described in Section 3.3.2 in the Section 3. In general, it appears that different drivers have different sets of enriched GO terms and BPWs associated with them, barring few exceptions. Moreover, for individual DAs, terms enriched among upregulated mRNAs are distinct from those associated with downregulated mRNAs. The distinctiveness underscores the fact that different DAs modulate the transcriptome and methylome in different ways to achieve their phenotypic goals.

### 4.8. Analyses of the DEMs and DMGs That Are Common between the DEMs and DMGs Associated with DAs in CDKN2A, EGFR and TP53

Individually, each driver led to distinctive sets of DEMs and DMGs with fairly distinct sets of enriched GO terms, as discussed above. However, the presence of some common elements between the lists revealed some interesting insights into potential co-regulatory mechanisms of these DAs on the transcriptome.

Several genes were found to be concurrently upregulated (and downregulated) in association with DAs in *CDKN2A* and *EGFR*. However, none were concurrently upregulated in association with DAs in *CDKN2A* and downregulated in association with *EGFR* (or vice versa). Thus, DAs in *CDKN2A* and EGFR have a cooperative effect in the co-regulation of the transcriptome from the perspective of these mRNAs. On the other hand, the transcriptional co-regulatory effect between either of these two (i.e., *CDKN2A* or *EGFR*) and *TP53* appears to be antagonistic. This is because several genes upregulated in association with DAs in *CDKN2A* [or *EGFR*] were concurrently downregulated in association with DAs in *TP53* (and vice versa).

## 5. Conclusions

Our main objective was to determine how DAs in common driver genes modulate the transcriptome, methylome and proteome in IDH1-W-GBs by examining the DEMs, DMGs and DEPs associated with them using Cbioportal. We observed that DAs in PTEN are unique because, unlike other DAs, they are unassociated with DEMs or DMGs. Different DAs are known to affect the transcriptome and methylome differently. However, such a complete lack of association with DEMs and DMGs has not been previously reported in association with any other DA or tumor, to the best of our knowledge. Moreover, *PTEN’s* lack of association with DEMs and DMGs is unaffected by concurrent DAs in other common drivers or by the types of DAs in *PTEN*. These findings indicate the ability of DAs in *PTEN* to get established within the heterogenous tumor landscape of IDH1-W-GBs without affecting its transcriptome or methylome. We showed that DAs in other drivers, such as CDKN2A, EGFR and TP53, are associated with fairly distinct sets of DEMs and DMGs with distinct contributions toward tumor phenotype, as apparent from the enriched GO terms and pathways. However, some common DEMs were noted among some of the sets. Analyzing these DEMs revealed some potential transcriptional co-regulatory mechanisms associated with these drivers. The transcriptional co-regulatory effects of the DAs in *CDKN2A* and *EGFR* were cooperative. In contrast, DAs in both *EGFR* and *CDKN2A* showed opposing transcriptional co-regulatory effects compared with DAs in *TP53*.

The insights obtained from the findings in our study can improve our understanding of the molecular mechanisms underlying gliomagenesis with potential therapeutic benefits.

## Figures and Tables

**Figure 1 brainsci-13-00186-f001:**
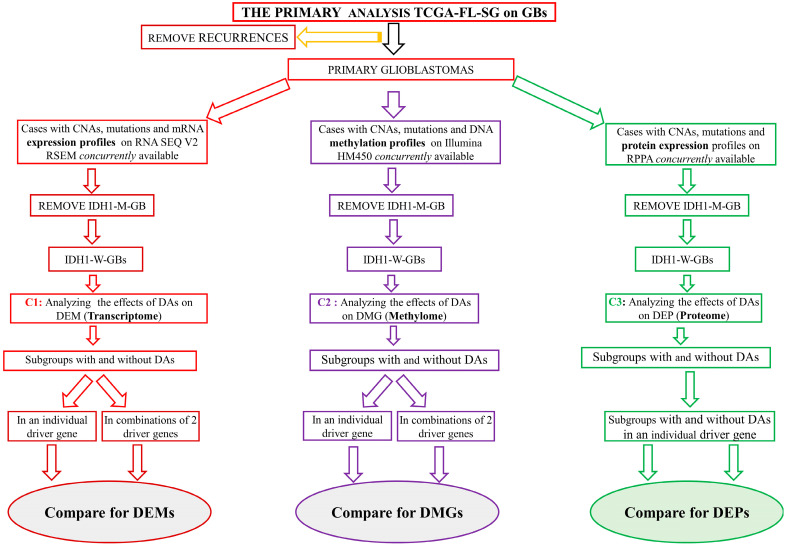
Workflow of the primary analyses performed on TCGA-FL-SG on GBs. Recurrent tumors were removed, and groups were created based on the availability of the required test profile to enable necessary analyses. IDH1-W-GBs were removed to create three core groups: C1, C2 and C3. The workflow for DEMs is indicated with a red color scheme, the one for DMG has a purple color scheme and the one for DEP has a green color scheme. TCGA-FL-SG: The Cancer Genome Atlas Study Group; GBs: Glioblastomas; IDH1-M-GB: IDH1 mutant glioblastoma; IDH1-W-GB: IDH1 wildtype glioblastoma; DA: Driver alteration; CNA: Copy number alteration; DEM: Differentially expressed mRNAs; DMG: Differentially methylated genes; DEP: Differentially expressed proteins; RNA seq v2 RSEM: RNA sequencing platform V2 and analyzed with RNA sequencing by Expectation Maximization (RSEM) software package (Version v 1.3).

**Figure 2 brainsci-13-00186-f002:**
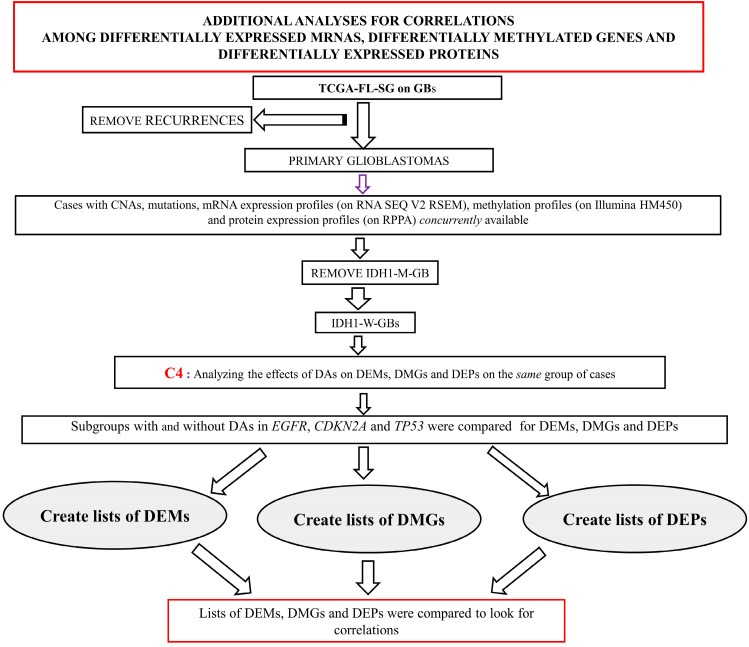
Workflow of the additional analyses performed on the C4 group where CNAs, mutations, mRNA expression profiles (on RNA SEQ V2 RSEM), methylation profiles (on Illumina HM450) and protein expression profiles (on RPPA) were concurrently available to determine correlations among DEMs, DMGs and DEPs. TCGA-FL-SG: The Cancer Genome Atlas Study Group; GBs: Glioblastomas; IDH1-M-GB: IDH1 mutant glioblastoma; IDH1-W-GB: IDH1 wildtype glioblastoma; DA: Driver alteration; CNA: Copy number alteration; DEM: Differentially expressed mRNAs; DMG: Differentially methylated genes; DEP: Differentially expressed proteins; RNA seq v2 RSEM: RNA sequencing platform V2 and analyzed with RNA sequencing by Expectation Maximization (RSEM) software package.

**Table 1 brainsci-13-00186-t001:** DAs in PTEN on DEMs and DMGs; groups of cases with combinations of appropriate DAs were compared.

Concurrent DAs in Other Driver Genes * along with DAs in *PTEN*	Groups Compared to Determine the Influence of the Presence or Absence of Other Concurrent DAs on the Non-Association of DAs in PTEN with DEMs and DMGs
DAs in *CDKN2A* present	CDKN2A + PTEN + vs. CDKN2A + PTEN-
DAs in *CDKN2A* absent	CDKN2A—PTEN + vs. CDKN2A—PTEN-
DAs in *EGFR* present	EGFR + PTEN + vs. EGFR + PTEN-
DAs in *EGFR* absent	EGFR—PTEN + vs. EGFR—PTEN-
DAs in *TP53* present	TP53 + PTEN + vs. TP53 + PTEN-
DAs in *TP53* absent	TP53—PTEN + vs. TP53—PTEN-
DAs in *RB1* present	RB1 + PTEN + vs. RB1 + PTEN-
DAs in *RB1* are absent	RB1—PTEN + vs. RB1—PTEN-

* Only four of the eleven other driver genes were selected for the analyses for reasons described above. DA: Driver alteration; DEM: Differentially expressed mRNA; DMG: Differentially methylated gene.

**Table 2 brainsci-13-00186-t002:** The prevalence of cases with driver alterations in 12 common driver genes along with their associated counts of differentially expressed mRNAs (upregulated, downregulated and total counts). The analyses were performed on the C1 group.

Genes with DAs	Number of Cases with DAs and the Corresponding Prevalence in Group C1	Number of Upregulated mRNAs in Group with DAs Compared to Group without DAs	Number of Downregulated mRNAs in Group with DAs Compared to Group without DAs	Total of Differentially Expressed mRNAs
*CDKN2A*	75 (58.13%)	81	221	302
*EGFR*	75 (58.13%)	613	664	1277
** *PTEN* **	**52 (40.31%)**	**0**	**0**	**0**
*TP53*	38 (29.45%)	1000	285	1285
*CDK4*	23 (17.83%)	794	722	1516
*PDGFRA*	20 (15.50%)	28	4	32
*MDM4*	12 (9.30%)	11	7	18
*RB1*	14 (10.85%)	94	76	170
*PIK3CA*	9 (6.97%)	4	9	13
*NF1*	12 (9.30%)	85	152	237
*MDM2*	9 (6.97%)	124	176	300
*PIK3R1*	6 (4.65%)	19	18	37

DA: Driver alteration.

**Table 3 brainsci-13-00186-t003:** Prevalence of cases with driver alterations in 12 common driver genes along with their associated counts of differentially methylated genes (hypermethylated, hypomethylated and total counts). The analyses were performed on C2 group.

Genes with DAs	Number of Cases with DAs and the Corresponding Prevalence in Group C2	Number of Hypermethylated Genes in Group with DAs Compared to Group without DAs	Number of Hypomethylated Genes in Group with DAs Compared to Group without DAs	Total of Differentially Methylated Genes
*CDKN2A*	67 (65.68%)	1	2	3
*EGFR*	44 (43.13%)	34	9	43
** *PTEN* **	**32 (31.37%)**	**0**	**0**	**0**
*TP53*	22 (21.56%)	2	22	24
*CDK4*	16 (15.68%)	9	3	12
*PDGFRA*	14 (13.72%)	1	18	19
*MDM4*	9 (8.82%)	0	53	53
*RB1*	8 (7.84%)	208	61	269
*PIK3CA*	12 (11.76%)	0	7	7
*NF1*	16 (15.68%)	639	135	774
*MDM2*	11 (10.78%)	10	0	10
*PIK3R1*	15 (14.70%)	0	0	0

DA: Driver alteration.

**Table 4 brainsci-13-00186-t004:** Prevalence of cases with driver alterations in 12 common driver genes along with their associated counts of differentially expressed proteins (overexpressed, underexpressed and total counts). The analyses were performed on the C3 group.

Genes with DAs	Number of Cases with DAs and the Corresponding Prevalence in Group C3	Number of Overexpressed Proteins in Group with DAs Compared to Group without DAs	Number of Underexpressed Proteins in Group with DAs Compared to Group without DAs	Total Differentially Expressed Proteins
*CDKN2A*	89 (67.42%)	1	0	1
*EGFR*	71 (53.78%)	8	6	14
*PTEN*	44 (33.33%)	6	0	6
*TP53*	37 (28.03%)	0	3	3
*CDK4*	22 (16.67%)	0	0	0
*PDGFRA*	19 (14.39%)	0	0	0
*PIK3R1*	15 (11.36%)	0	0	0
*PIK3CA*	11 (8.33%)	0	2	2
*MDM2*	11 (8.33%)	0	0	0
*MDM4*	13 (9.84%)	0	0	0
*NF1*	17 (12.87%)	1	4	5
*RB1*	12 (9.09%)	0	0	0

DA: Driver alteration.

**Table 5 brainsci-13-00186-t005:** The influence of the presence or absence of concurrent DAs in four driver genes (other than *PTEN*) on the associations between DAs in *PTEN* with DEMs and DMGs. The numbers in round parentheses adjacent to a group represent the number of cases of that group in C1 that were used to compare differentially expressed mRNA. The numbers in square parentheses adjacent to a group represent the number of cases of that group in C2 that were used to compare differentially methylated genes.

Groups Compared	Upregulated mRNA Count	Downregulated mRNA Count	Hypermethylated Gene Count	Hypomethylated Gene Count
*CDKN2A+PTEN+ **(29)/*** **[19]** **vs.** *CDKN2A+PTEN- **(46)/*** **[48]**	0	0	0	0
*CDKN2A-PTEN+ **(23)/*** **[13]** **vs.** *CDKN2A-PTEN- **(31)/*** **[22]**	0	0	0	0
*EGFR+PTEN+ **(28)/*** **[13]** **vs.** *EGFR+PTEN- **(47)/*** **[31]**	0	0	0	0
*EGFR-PTEN+ **(24)/*** **[19]** **vs.** *EGFR-PTEN- **(30)/*** **[39]**	0	0	0	0
*TP53+PTEN+ **(16)/*** **[7]** **vs.** *TP53+PTEN- **(22)/*** **[15]**	0	0	0	0
*TP53-PTEN+ **(36)/*** **[25]** **vs.** *TP53-PTEN- **(55)/*** **[55]**	0	0	0	0
*RB1+PTEN+ **(10)/*** **[4]** **vs.** *RB1+PTEN- **(4)/*** **[4]**	0	1	0	0
*RB1-PTEN+ **(42)/*** **[28]** **vs.** *RB1- PTEN- **(73)/*** **[62]**	0	0	0	0

## Data Availability

This study was performed using data that were downloaded and analyzed on a publicly available web portal, i.e., Cbioportal, which hosts data from publicly available databases such as “The Cancer Genomic Atlas”. The relevant lists of differentially expressed genes and differentially methylated genes between different groups of tumors derived from the above public resources are provided as Appendix A. If the editors or reviewers require any other data relevant to the study, such data can be submitted for review and publication if deemed necessary.

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
