# Peer review of "Non-Association of Driver Alterations in PTEN with Differential Gene Expression and Gene Methylation in IDH1 Wildtype Glioblastomas"

_brainsci, 2023, doi:10.3390/brainsci13020186_

Round 1

Reviewer 1 Report (Previous Reviewer 3)

In the current manuscript, the authors have improved a lot according to the reviewer’s suggestion. I have only several minor points.

1.       The authors include an additional analysis named C4 group in the method, however, in figure 1, they did not include this new analysis.

2.       Table 6 and Table 7 are exceptionally large, and it is not easy for the readers to see the whole tables. Could the authors move these 2 tables into supplementary tables shown in Excel files?

3.       Line 574: the symbol dot is far away from the number “4.2”.

Author Response

Please read the attachment, thank you.

Reviewer 2 Report (New Reviewer)

Here, Mallik et al. found non-association of driver alterations in the tumor suppressor gene PTEN at the level of mRNA and gene methylation in IDH1-wildtype-glioblastomas. Their data resource, 12 driver gene candidates, experimental design, results and conclusions are well process and made. 

Author Response

Please read the attachment, thank you.

Reviewer 3 Report (New Reviewer)

Reviewer's Comments on Manuscript-Brainsci-2110139-Peer-Review-v1

The manuscript entitled Non-Association of Driver Alterations in PTEN with Differential Gene Expression and Gene Methylation in IDH1-wildtype-Glioblastomas is based on the study of analyzing the effect of DAs in common driver genes modulate 723 the transcriptome, methylome and proteome in IDH1-W-GBs by examining the DEMs, 724 DMGs and DEPs associated with them using Cbioportal.

The manuscript has been written well. The introduction part is based on both old and new literature related to this work. The experimental method used is highly efficient in producing the expected results. The “result and discussion” part is quite comprehensive. The supplementary data provided fully justify and verify these given results. The references provided fully support the given text. 

Author Response

Please read the attachment, thank you.

This manuscript is a resubmission of an earlier submission. The following is a list of the peer review reports and author responses from that submission.

Round 1

Reviewer 1 Report

Major issues:

- as the authors write in the introduction, driver alterations are have the potential to drive the neoplastic process. Considering that the significance of most mutations in cancer is unknown, it is unclear how the authors selected driver alterations. Since this is the focus of the paper, this can not be evaluated in the current form. The authors say that cBioportal "identifies candidate driver gene using the MutSig algorithm " (line 169). The reference given is out of place. In addition, the MutSig algorithm does not identify driver genes, but identifies genes that were mutated more often than expected by chance given background mutation processes. These have the potential to be driver genes. Mutations and CNA to these genes have the potential to be driver events. For any given genes, cBioportal offers the possibility to query for driver mutations and CNA, but it doesn't look like that the authors included this filter. Overall, the methods are not clear, and it is not possible for the reader to understand and replicate the analysis that the authors performed. Methods must be presented in a way that the analyses can be reproduced by the reader.

- in many sections of the manuscript the authors write that by analyzing differentially expressed genes/proteins of methylome signatures in different subsets of tumors, they are analyzing the EFFECT of driver genes. This is a misconception. The effect of driver genes or mutations/CNA can only be assessed experimentally. In their analysis the authors are looking at associations, most of which are likely incidental findings with limited biological significance.

- Most of the references are out of place, or not related to the text. These issues occur throughout the text, but here a few examples:

lines 38-39: the reference is about targeting of EGFR, not on different mutations, so out of place. The research article describing the mutations are not referenced.

lines 41-43: the references are not related top the manuscript text.

line 45: reference out of place.

Minor issues

line 38: abbreviation not defined

Sometimes "glioblastoma" is abbreviated as "GB", sometimes no. This needs to be harmonized.

English readability: lines 45-47, lines 52-54, lines 56-60

Author Response

Please see attachment for author's reply.

Reviewer 2 Report

In the present work, the Authors reconsider data in CBioportal to examine the effect of  multiple driver mutations on DNA methylation, gene expression and protein expression in an assembled cohort oh IDH-wildtype gliobastomas.

The work is introduced by a an extensive definition and description of driver mutations and whether these alterations can affect the molecular profile of tumor cells and, therefore, impact the clinical aspects associated with the pathology. This introduction is clear and provide the reader with sufficient elements to appreciate the importance of investigating the downstream effect of cancer mutations. However, the introduction seems to me too extensive, as it should be simply intended  to introduce the scientific question for a scientific audience that, I suppose, will be quite specialized. As an example, at line 68 Authors state that “Of note, the earlier Roman numerals of WHO grading have been replaced by Arabic numerals in the 2021 WHO classification of CNS tumors”, which doesn’t even need to be mentioned in my opinion. Nonetheless, Authors then use these “Arabic numerals” in the discussion (see line 438).

Differential DNA methylation and expression associated with the 4 most-recurrent alterations in IDH-wt GB are presented (as extended data).  Interestingly, the Authors observe that PTEN alterations alone do not seem to associate with any DNA methylation or transcriptional changes in tumors. Few  changes in the proteomic profiles are detected as associated with PTEN alterations.

Overall, I found the scientific hypotheses behind this work, as well as the questions meant to be answered quite unclear. What are the Authors trying to add to our current knowledge, precisely? All driver mutations have, by definition, an impact on the tumor cell phenotype, independently on whether they induce an epigenetic and transcriptional remodeling or not. I agree that the findings presented on PTEN might represent an exception to a common caveat, but this observation alone does not constitute to me a significative advance.

-       As a tumor biologist, I believe that the work will benefit of a deeper investigation of the functional aspects related to the changes associated with different driver mutations. E.g. what do the list of DMG and DEM tell us about the role of CDKN2A, EGFR, or TP53 mutations compared to PTEN, functionally? What GO or pathways are enriched in the profiles of these alterations and are missing in PTEN-mutated GB? How do alterations in these pathways or molecular functions may impact the clinical outcomes of patients in different manners? I hope the Authors will agree that a functional investigation of their observation will make the conclusions less speculative.

-       At line 459, Authors state that “It would have been ideal to analyze mRNA expression, gene methylation and protein expression on the same group... Instead, creating three separate groups, enabled us to perform our analyses on considerably larger groups, each with more than 100 cases”. I do not agree with their conclusion. Statistical significance indicates that alterations in gene methylation and expression are frequent in the cohorts displaying the driver alteration compared to WT tumors. It would be interesting to know whether DNA methylation patterns correlates with changes in mRNA and protein expression profiles, even if the size of the cohorts is not identical for the different datasets. Otherwise, why considering the three types of profiles instead of focusing on one of them?  

-       Also, the discussion is too long compared to the amount of new data presented, and often tends to justify the design of the study. I think this part can simplified by avoiding explanations that are redundant with the text. As mentioned above, many conclusions are at the moment purely speculative and will need further investigation on the functional aspects to support them.

Minor points :

-       In general, many sentences are quite complex and difficult to follow. Authors should shorten certain sentences or made them more fluent.

-       Line 100: “the later” should be replaced by “the latter”.

-       Line 137: the layout of the abbreviation section after figure 1 seems bizarre.

-       At line 314 there is a repetition of “EGFR”. I think the second EGFR in the sentence should be replace by TP53.

Author Response

(The authors gave the same response as above.)

Reviewer 3 Report

In this work, Mallik et al. aimed to figure out the effects of driver alterations (DAs) on differentially expressed mRNAs (DEMs), differentially methylated genes (DMGs), and differentially expressed proteins (DEPs) in Isocitrate Dehydrogenase 1 wildtype glioblastomas (IDH1-W-GBs). They used Cbioportal to analyze the 12 top DAs in IDH1-W-GBs and found DAs in PTEN are not related to any DEMs or DMGs. In general, the analysis method is quite simple, the data is very less and the conclusion is not solid as well. The detailed comments are as follows:

1.  The authors showed DAs in PTEN are not associated with DEMs, which is an unexpected find. Besides DEMs, the changes in gene copy numbers are also very common in cancers, are DAs in PTEN related to gene copy number change in IDH1-W-GBs?

2. Based on TCGA data on cBioPortial the authors used, the DNA methylation (HM450) and protein expression (RPPA) data only cover a small portion of samples, so the conclusion based on these data may not be that solid.

3. For the data shown in Table 5, almost all the groups showed no changes in DEMs and DMGs. For each group, how many samples are included in each group? Is the sample number in each group sufficient for comparison?

4.  For the 3.2 parts, please show the data the authors analyzed from Brain tumor PDX (Mayo Clinic 2019).

5. Figure 2 is not in a figure format, please delete and describe the conclusion in the context.

Author Response

Please see attachment for author' reply.

Round 2

Reviewer 2 Report

None of the major concerns raised during the first stage of reviewing has really been addressed by the Authors.

E.g. in the comment number 4, Authors were asked to perform GO or similar enrichment analysis, which does not require out of topic skill and should not represent an excessive amount of work. In their answer, Authors claim that "including these findings would considerably increase the length of the manuscript", and that they "intend to share these results through future publications". I don't consider such a response as acceptable. It is our duty, as scientists, to present our findings and respond to colleagues' concerns in the most accurate and concise way. The length of multiple sections of this manuscript is already excessive in the present form, although the scientific advance presented is limited.

The revision of the work has only consisted in minor corrections according to suggestions.

I would like to stress the fact that revision after peer-reviewing does not merely represent a formality. Rather, revision  is a key process allowing to ameliorate the quality and the clarity of the presentation of scientific findings.

I do not consider that the manuscript has been sufficiently improved, and I therefore cannot express a favorable recommendation.

Author Response

Dear Reviewer,

Please read the attachment for detailed response.

Thank you.

Reviewer 3 Report

For the second revision, the authors answered all my questions and improved the manuscript format.  I have no further questions.

Author Response

Dear Reviewer,

Thank you very much!